# The Spatiotemporal Variations in Soil Erosion and Its Dominant Influencing Factors in the Wenchuan Earthquake-Stricken Area

Jialin Li [1,†], Bing Guo [1,2,*,†] , Guang Yang [2,*] and Kun Yu [2]

1 School of Civil Engineering and Geomatics, Shandong University of Technology, Zibo 255000, China; 21507020776@stumail.sdut.edu.cn

2 Research Institute of Aerospace Information, Chinese Academy of Sciences, Beijing 100101, China; yukun@aircas.ac.cn

* Correspondence: guobing@sdut.edu.cn (B.G.); yangguang@aircas.ac.cn (G.Y.); Tel.: +86-1876-696-5987 (B.G.); +86-1851-058-6581 (G.Y.)

† These authors contributed equally to this work.

**Abstract:** Earthquakes have obvious influences on the spatiotemporal changes in soil erosion intensity in earthquake-stricken areas. However, fewer studies have been conducted to evaluate the spatiotemporal changes in soil erosion before and after the Wenchuan earthquake and its dominant factors in different periods. In order to explore the above issue, this study quantitatively analyzed the spatiotemporal variation characteristics of soil erosion in the Wenchuan earthquake-stricken area from 2000 to 2019 based on the RUSLE model, gravity center model, and its dominant factors in different periods were determined using Geodetector. The research results indicated that: (1) The amount of mean total erosion in the Wenchuan earthquake-stricken area during 2000–2019 was $10.05 \times 10^8$ t, with an average soil erosion modulus of 2038.2 t/(km²·a), indicating mild erosion. (2) The spatiotemporal patterns of soil erosion changed greatly in the Wenchuan earthquake-stricken areas during 2000–2019. Areas with intensified soil erosion were mainly distributed in Lixian, Wenchuan, Xiaojin, and other areas near the Longmenshan fault zones. (3) Landslides, debris flows, and floods caused by the Wenchuan earthquake contributed to aggravating the soil erosion intensity in the stricken area. (4) During 2000–2019, the soil erosion intensity showed an overall decreasing trend, while the soil erosion intensity showed an increasing trend around 2008 due to the Wenchuan earthquake. (5) During 2000–2019, soil erosion in the Wenchuan earthquake-stricken area has been greatly affected by vegetation, terrain, and land use types. The research results could provide important decision-making support for soil erosion prevention and ecosystem restoration in the Wenchuan earthquake-stricken area. In addition, these results would be conducive to revealing and understanding the interactive process of "Water–Soil–Vegetation" in mountainous regions all over the world.

**Keywords:** soil erosion; Wenchuan earthquake; gravity center; spatiotemporal pattern; driving mechanism

## 1. Introduction

Soil erosion refers to the phenomenon of the destruction, separation, transportation, and deposition of soil and its parent materials on the surface of the earth under the influence of natural factors and human factors under the action of external forces such as water power, wind power, freeze–thaw cycles, and gravity [1]. Soil erosion destroys land resources, causing a large amount of fertile topsoil to be lost, rapidly reducing soil fertility and plant yield. Under specific geological conditions, soil erosion can also cause geological disasters such as landslides, collapses, and debris flows. The resulting soil erosion can cause river siltation and aggravated flood disasters [2]. Natural disasters such as earthquakes and debris flows have accelerated soil erosion, so soil erosion assessment in earthquake-stricken areas has become an important part of post-disaster reconstruction [3]. In addition,

revealing and clarifying the dominant factors of soil erosion in earthquake-stricken areas contributes to the accurate governance and restoration of the ecological environment.

The universal soil loss equation is an empirical soil erosion prediction model developed in the United States to quantitatively predict the average soil loss of farmland or grassland slopes. These studies mainly investigate the hydrodynamic soil erosion caused by rainfall [4]. Since the 1990s, a large number of researchers have studied the spatial and temporal patterns of soil erosion based on the Universal Soil Loss Equation (USLE) and its revised version, the Universal Soil Loss Equation (RUSLE). Compared with the USLE model, the RUSLE model can be applied in non-agricultural areas and considers more factors affecting soil erosion. Bircher et al. [5] found that the LS factor of RUSLE based on field block calculations provided many advantages in determining the channel network and maximum flow length. Fayas et al. [6] found that land use management practices were better reflected by the RUSLE. Chuenchum et al. [7] quantitatively assessed annual soil erosion in the future scenarios of 2030 and 2040 according to the spatial distribution and trend in sediment yield with regard to climate and land change. Liang et al. [8] calculated the soil erosion modulus in the Yan'an area in 2012, 2015, and 2018 and then explored the spatial and temporal changes in soil erosion in the Yan'an area. Sun et al. [9] quantitatively analyzed the spatial distribution characteristics of soil erosion in typical small watersheds in the middle reaches of the Yellow River based on the RUSLE model and determined the sensitive areas of soil erosion. Chen et al. [10] quantitatively evaluated and analyzed the spatiotemporal change in patterns of soil erosion in the Huangshui River Basin from 2000 to 2015 and revealed the variation characteristics of soil erosion. Luetzenburg et al. [11] applied the geospatial interface of the Soil Erosion Prediction Project (GeoWEPP) and RUSLE to evaluate the soil erosion intensity in two agricultural catchments with fine spatial resolution. However, at present, there are relatively few studies on the long-term temporal and spatial evolution characteristics of soil erosion in earthquake-stricken areas. Jiang et al. [12] analyzed the spatial distribution pattern of different erosion intensities in the Lushan earthquake-stricken area based on the RUSLE model. Xia et al. [13] used the USLE model to quantitatively analyze the spatial distribution characteristics of soil erosion under the earthquake effect in Jiuzhaigou. The Wenchuan earthquake in 2008 was the most destructive geological disaster since the founding of the People's Republic of China, which has caused a huge threat to the ecological environment in the stricken area. At the same time, it has caused secondary disasters such as debris flow, landslides, and collapses, which have aggravated soil erosion intensity in the earthquake-stricken area [14]. Moreover, the dominant factors relating to the changes in the process of soil erosion intensity in different periods (especially before and after the Wenchuan earthquake) are different and unclear, which must be urgently explored.

Based on the RUSLE model, gravity center model, and Geodetector, this study quantitatively analyzed the spatial and temporal variation characteristics of soil erosion in the Wenchuan earthquake-stricken area from 2000 to 2019 and revealed and clarified the dominant factors affecting the spatial and temporal changes in soil erosion in earthquake-stricken areas in different historical periods, which could provide important decision support for the prevention and control of soil erosion and the restoration of ecological environment in post-earthquake stricken area.

## 2. Materials and Methods

### 2.1. Study Area

The Wenchuan earthquake-stricken area (N27°30′~N36°56′, E100°38′~E108°58′, Figure 1) is located in Yingxiu Town, Wenchuan County, Aba Tibetan and Qiang Autonomous Prefecture, Sichuan Province, covering an area of $48.5 \times 10^4$ km². Among them, there are 10 counties (cities) in the extremely heavy stricken area, 41 counties (cities) in the relatively heavy stricken area, and 186 counties (cities) in the generally stricken area. The geological structure of the study area belongs to the Longmenshan fault zone. The mountains are widely distributed, and their altitude ranges from 184 m to 6261 m. The vertical zone

changes, obviously, and the climate types are diverse. The monsoon climate is formed by the influence of three circulations: Siberian westerly airflow, Indian Ocean warm current, and Pacific southeast monsoon. The seasonal distribution of precipitation is uneven, mainly concentrated from May to October, accounting for 80%~90% of the total annual precipitation [15]. Regarding the altitude from high to low, the soil types mainly include alpine desert soil, alpine meadow soil, and subalpine meadow soil distributed in zones with altitudes > 4000 m. The soil types distributed in zones with altitudes of 1500~3200 m are mainly brown, dark, and yellow–brown soil [16]. The soil types distributed in zones with altitude < 1500 m are mainly yellow soil and its subtypes. Affected by the earthquake, some of the surface soil with high vegetation coverage disappeared and became barren initial gravel soil [17]. Due to the destruction of secondary geological disasters such as earthquakes, landslides, and debris flows, the original vegetation in some areas has been destroyed. At present, artificial forests such as Cryptomeria fortunei, Cunninghamia lanceolata, Betula platyphylla, and Alnus cremastogyne are the main forest types [18].

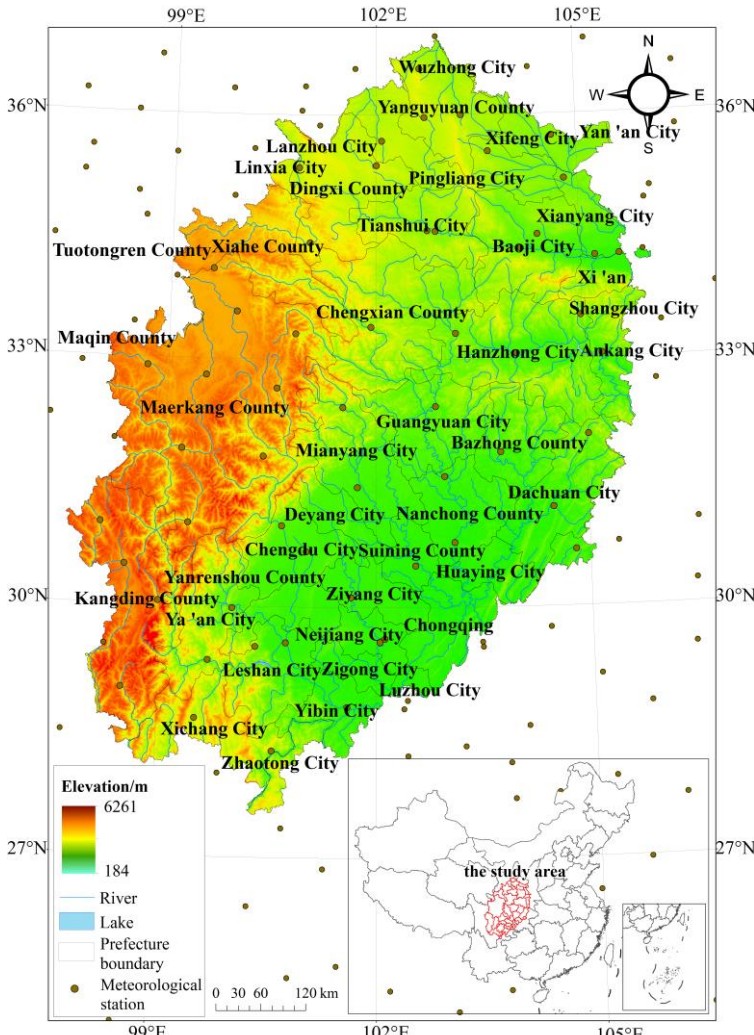

**Figure 1.** Location map of the research area.

### 2.2. Data Source and Preprocessing

The 2000–2019 NDVI datasets were derived from MOD13Q1 in NASA's LAADS DAAC website with spatiotemporal resolutions of 16 and 250 m, respectively. The MRT tool was used to convert and resample the above datasets (HDF → tif, 250 m → 100 m). DEM data were derived from the SRTM dataset with a spatial resolution of 90 m. Daily precipitation data in the Wenchuan earthquake-stricken area and surrounding meteorological stations

were collected by the China Meteorological Data Sharing Network. The 30 m resolution land use data for 2000, 2005, 2010, 2015, and 2020 were obtained from the Resources and Environmental Science and Data Center. Data from 1:100,000 soil types were obtained from the Nanjing Institute of Soil Science, Chinese Academy of Sciences. The MVC method was used to synthesize a maximum value of 16-day NDVI data to obtain the maximum NDVI data from 2000 to 2019. Population density, GDP data, and disaster vector datasets for Sichuan Province were collected by the Resources and Environmental Science and Data Center. All the above datasets were unified into Krasovsky Albers equal-area projection and raster data output pixel sizes of 100 m using ArcGIS 10.7 (Environmental Systems Research Institute, Inc., RedLands, CA, USA) reprojection and resampling tools.

*2.3. Methods*

2.3.1. Soil Loss Equation of Earthquake-Stricken Area

The modified universal soil loss equation (RUSLE) was used to calculate soil erosion in the Wenchuan earthquake-stricken area [6]. Its equation is as follows:

$$A = R \times K \times LS \times C \times P \tag{1}$$

where $A$ is the soil erosion modulus [t/(km$^2$·a)]; $R$ is the rainfall erosivity factor [(MJ·mm)/(km$^2$·h·a)]; $K$ is soil erodibility factor [(t·km$^2$·h)/(km$^2$·MJ·mm)]; $LS$ is the slope aspect factor; $C$ is vegetation cover factor; $P$ is soil and water conservation measure factor, including engineering measure and farming measure factor.

(1)    *R* factor

Rainfall erosivity represents the dynamic index of soil erosion caused by rainfall, reflecting the potential capacity of soil erosion caused by rainfall, and its magnitude is related to rainfall and rainfall intensity [19]. The rainfall erosivity factor is the primary basic factor in the RUSLE model. In this study, according to Liu et al. [20], the calculation formula of the rainfall erosivity estimation model was improved. The *R* factor was obtained based on daily precipitation using the tools of *C#* and interpolation in ArcGIS 10.7. The formula is as follows:

$$\overline{R_k} = \frac{1}{N}\sum_{i=1}^{N}\left(\alpha\sum_{j=1}^{M}P_{d_{ikj}}^{\beta}\right) \tag{2}$$

$$\alpha = 21.239\beta^{-7.3967} \tag{3}$$

$$\beta = 0.6243 + \frac{27.346}{\overline{P_{d_0}}} \tag{4}$$

$$\overline{P_{d_0}} = \frac{1}{N}\sum_{i=1}^{N}\sum_{k=1}^{12}\sum_{j=1}^{M}P_{d_{ikj}} \tag{5}$$

$$\overline{R} = \sum_{k=1}^{12}\overline{R_k} \tag{6}$$

where $\overline{R_k}$ is the rainfall erosivity of the kth month (MJ·mm/km$^2$·h); $N$ is the length of the calculated data series; $M$ is the number of erosive rainfall in the kth month of the ith year; $P_{d_{ikj}}$ is the jth erosion in the $k$th month of the ith year, and the daily rainfall $\geq$12 mm is regarded as erosive rainfall. $\alpha$ and $\beta$ are model parameters, which need to be estimated using Equations (3)–(5). $\overline{P_{d_0}}$ is the annual mean value of erosive rainfall (mm). $\overline{R}$ is the average annual precipitation erosivity [(MJ·mm/(km$^2$·h·a))].

(2)    Soil erodibility factor *K*

Soil erodibility is the characterization of soil infiltration capacity to rainfall, and its sensitivity to rainfall and runoff erosion and transport, and it is an intrinsically important

factor affecting soil loss. The value of *K* is mainly related to soil texture and organic matter content [21]. The Integrated Climate Model for Environmental Policy (EPIC) is one of the methods used to calculate soil erodibility, which can be obtained from the soil database and ArcGIS 10.7. Its formula is:

$$
\begin{aligned}
K = &\{0.2 + 0.3\exp[-0.0256S_d(1 - S_i/100)]\} \times \\
&[S_i/(C_i + S_i)]^{0.3} \times \\
&\{1 - 0.25C/[C + \exp(3.72 - 2.95C)]\} \times \\
&\{1 - 0.007S_d/[0.007S_d + \exp(-5.51 + 0.229S_d)]\}
\end{aligned}
\tag{7}
$$

where $S_d$ is the percentage content of sand particles (particle size 0.05~2 mm); $S_i$ is the percentage content of silt (particle size 0.002~0.05 mm); $C_i$ is the percentage of clay particles (particle size < 0.002 mm); *C* is the percentage of organic carbon.

(3)　Slope length factor LS

Topography and geomorphology are important factors affecting the layout of soil erosion and soil and water conservation measures. The larger the slope and the longer the slope, the greater the runoff energy and runoff volume and the stronger the erosion effect [22]. Its calculation formula is as follows:

$$
S = \begin{cases}
10.8\sin\theta + 0.03 & \theta < 5° \\
16.8\sin\theta - 0.5 & 5° < \theta \le 10° \\
20.204\sin\theta - 1.2404 & 10° < \theta \le 25° \\
29.585\sin\theta - 5.6079 & \theta > 25°
\end{cases}
\tag{8}
$$

The calculation formula of slope length factor *L* is as follows:

$$
L = (\lambda/22.13)^m
$$
$$
m = \begin{cases}
0.2, \theta \le 1° \\
0.3, 1° < \theta \le 3° \\
0.4, 3° < \theta \le 5° \\
0.5, \theta > 5°
\end{cases}
\tag{9}
$$

where *S* is the slope factor (dimensionless), $\theta$ is the slope (°); *L* is the slope length factor; $\lambda$ is the slope length (m). The calculation of *LS* is conducted using the Raster Calculator in ArcGIS 10.7.

(4)　Vegetation cover factor *C*

The vegetation cover factor is an important inhibiting factor in erosion dynamics, closely related to vegetation cover degree [18]. It represents the ratio of soil and water loss between surfaces with high vegetation coverage and completely barren surfaces under the same conditions, and its value is between 0 and 1 [23]. Generally speaking, the higher the vegetation coverage, the more obvious the effect of soil and water conservation. According to the research method of Cai et al. [24], vegetation cover factor *C* can be expressed as:

$$
C = \begin{cases}
1, f_c = 0 \\
0.6508 - 0.3436\lg f_c, 0 < f_c < 0.783 \\
0, f_c \ge 0.783
\end{cases}
\tag{10}
$$

$$
f_c = \frac{\text{NDVI} - \text{NDVI}_{\text{soil}}}{\text{NDVI}_{\text{max}} - \text{NDVI}_{\text{soil}}}
\tag{11}
$$

where *C* refers to the vegetation cover factor, $f_c$ is the vegetation coverage $\text{NDVI}_{\text{soil}}$ is the NDVI value of barren land because different soil attributes are different, and its value ranges from $-0.2$~0.1 [23]. NDVI max refers to the NDVI value in a fully vegetation-covered region. In this study, this process was conducted using the Raster Calculator Tool in ArcGIS 10.7.

(5) Soil and water conservation measures factor $P$

The soil and water conservation measure factor refers to the ratio of soil loss after soil and water conservation measures are taken to soil loss during downhill planting, which represents the impact of crop management measures on soil loss, and its value is between 0 and 1. The value of land use type with no water conservation measures is 1, and the value of land use type with little soil erosion is 0. In this study, the $p$-values of different land use types were obtained according to the land uses of stricken areas and the previous research results of related research methods. Among them, the $p$-value of forested land, high coverage meadow, medium coverage meadow, construction land, and unused land was assigned a value of 1, the $p$-value of low coverage meadow was 0.7, the $p$-value of dry land was 0.4, the $p$-value of paddy fields was 0.15, and the $p$-value of water areas was 0.

### 2.3.2. Soil Erosion Intensity Index

In order to analyze the relationship between different land use types and soil erosion, the soil erosion intensity index was adopted in this study to reflect the fact that it takes into account the area occupied by each land use type [15]. This index can be obtained using the Zonal Statistics Tool in ArcGIS 10.7, and the formula is:

$$E_j = 100 \times \sum_{i=1}^{n} C_i \times A_{ij} \tag{12}$$

where $E_j$ is the soil erosion intensity index of the JTH land use mode; $C_i$ is the classification value of soil erosion intensity in Class i of the i land use mode. $A_{ij}$ is the percentage of area occupied by Type i soil erosion in type j land use mode. In this study, the values of soil erosion intensity from weak to strong were 1, 2, 3, 4, 5, and 6.

### 2.3.3. Gravity Center Model

A gravity center can represent the spatial and temporal distribution of geographical elements and is widely used in the fields of economics, population studies, ecology, etc. In the field of soil erosion, the spatial variation characteristics of a gravity center can reflect the changing trend and its differences in soil erosion in the study region. The gravity center of soil erosion can be obtained using ArcGIS 10.7. The erosion value of grid $i$ is $v_i$, and the gravity center calculation formula of soil erosion can be expressed as follows:

$$\overline{x} = \frac{\sum_{i=1}^{n} v_i x_i}{\sum_{i=1}^{n} v_i} \tag{13}$$

$$\overline{y} = \frac{\sum_{i=1}^{n} v_i y_i}{\sum_{i=1}^{n} v_i} \tag{14}$$

### 2.3.4. Geodetector

Geodetector is an effective tool for detecting spatial differentiation. The spatial differentiation of soil erosion in earthquake-stricken areas was detected using the Geodetector model to reveal the explanatory power of a certain factor to soil erosion. The expressions are as follows:

$$q = 1 - \frac{\sum_{h=1}^{L} N_h \sigma_h^2}{N \sigma^2} = 1 - \frac{SSW}{SST} \tag{15}$$

$$SSW = \sum_{h=1}^{L} N_h \sigma_h^2, \ SST = N\sigma^2 \tag{16}$$

where $h = 1, \ldots, L$ is the stratification of independent variable $Y$ or dependent variable $X$, that is, classification or partition; $N_h$ and $N$ are the number of units in layer $h$ and the whole region, respectively. $\sigma_h^2$ and $\sigma^2$ are the variances of $Y$ values for layer $h$ and the whole area, respectively. SSW and SST are the sum of intra-layer variance and the total variance in the whole region, respectively.

## 3. Results

### 3.1. Spatial Distribution Pattern of Soil Erosion from 2000 to 2019

Based on ArcGIS 10.7 and the RUSLE model, the soil erosion time series dataset of the Wenchuan earthquake-stricken area from 2000 to 2020 was obtained. The soil erosion modulus was graded according to the SL190-2007 soil erosion classification and classification standard (Table 1), and the distribution of average soil erosion intensity before and after the Wenchuan earthquake in the preceding 20 years (Figure 2) and the area of soil erosion at different levels (Table 2) was obtained.

**Table 1.** Classification of soil erosion intensity.

| Classification of Erosion Intensity | Erosion Modulus/t·(km²·a)$^{-1}$ |
| :---: | :---: |
| micro-erosion | <500 |
| mild erosion | 500~2500 |
| moderate erosion | 2500~5000 |
| intensive erosion | 5000~8000 |
| extreme erosion | 8000~15,000 |
| severe erosion | >15,000 |

As shown in Table 2 and Figure 2, the average total erosion amount in the Wenchuan earthquake-stricken area was $10.05 \times 10^8$ t/a during 2000–2019, and the average soil erosion coefficient was 2038.2 t/(km·a)$^{-1}$, which indicated mild erosion. Among them, the areas of micro-erosion were the largest, which was $37.64 \times 10^4$ km², accounting for 76.34% of the total area. The total amount of erosion accounted for only 0.88%, which was mainly distributed in Mianyang City, Guangyuan City and Bazhong City, Hanzhong City, Shangzhou City, Ankang City, Chengdu City, and Ya'an City. The areas of moderate erosion zone totaled $2.94 \times 10^4$ km², accounting for 5.96% of the total erosion area, which was mainly distributed in Dingxi County, Guyuan County and Pingliang City, Nanzhong County, Mianyang City, and Cheng County. The areas of intensive erosion, extreme erosion, and severe erosion were relatively small, with erosion areas of $1.86 \times 10^4$ km², $1.84 \times 10^4$ km², and $1.67 \times 10^4$ km², accounting for 3.76%, 3.74%, and 3.18% of the total area, respectively, and the sum of total erosion amounts accounted for 10.88%. These areas were mainly distributed to the west of Longmen toward Sanmenxia, the north of Mintuo River basin, and the east of Shigu River basin below Jinsha River, such as Yan'an City, Songpan County, Heishui County, Lixian County, Wenchuan County, Luding County, and Kangding City.

**Table 2.** Average soil erosion intensity classification in the Wenchuan earthquake area from 2000 to 2019.

| Erosion Intensity | Erosion Area/ $10^4$ km$^2$ | Erosion Modulus/ (t·(km$^2$·a)$^{-1}$) | Total Erosion/ (10,000 t·a$^{-1}$) | Area Ratio/% | Erosion Rate/% |
|---|---|---|---|---|---|
| micro-erosion | 37.64 | 23.01 | 866.43 | 76.34 | 0.88 |
| mild erosion | 3.36 | 1568.71 | 5230.14 | 6.81 | 5.24 |
| moderate erosion | 2.94 | 3620.94 | 10,643.76 | 5.96 | 10.52 |
| intensive erosion | 1.86 | 6336.11 | 11,759.99 | 3.76 | 11.6 |
| extreme erosion | 1.84 | 10,903.37 | 20,102.10 | 3.74 | 19.9 |
| severe erosion | 1.67 | 31,173.59 | 51,880.38 | 3.38 | 51.88 |

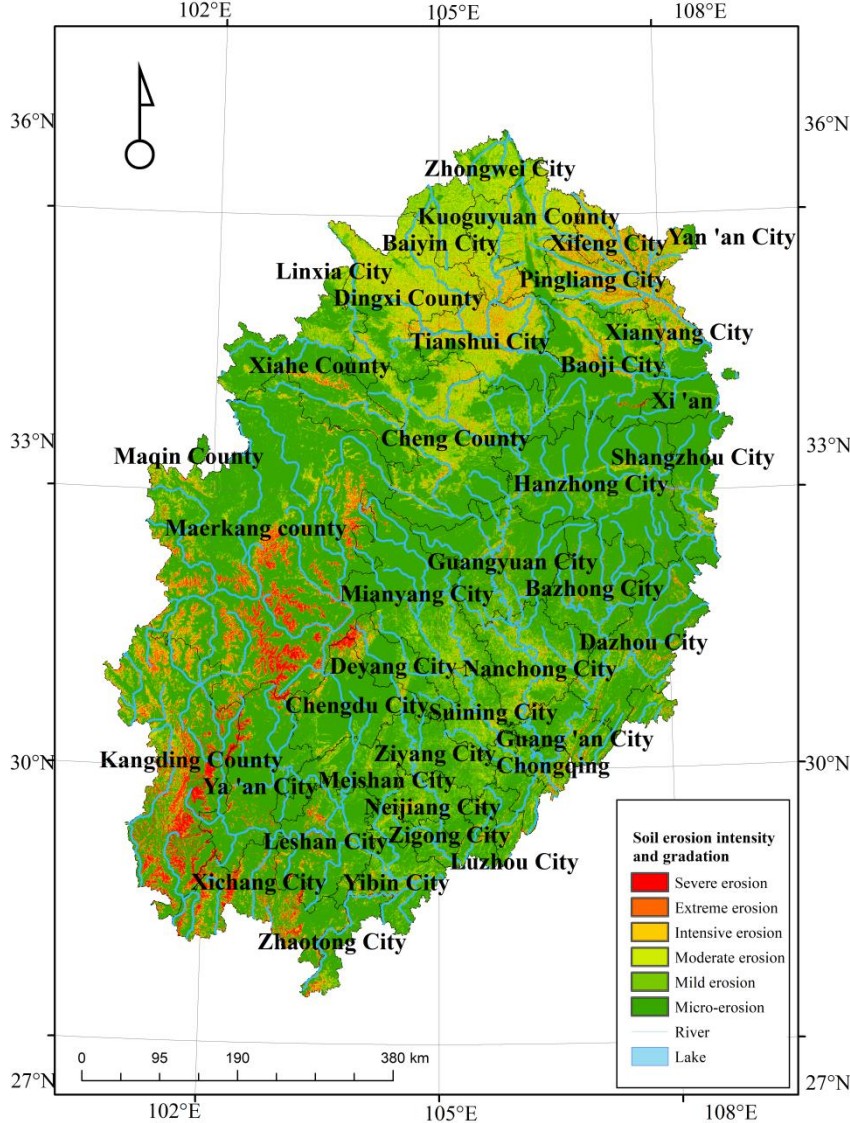

**Figure 2.** Average soil erosion intensity in the Wenchuan earthquake-stricken area from 2000 to 2019.

As shown in Figure 3, the proportional area of mild erosion showed an increasing trend, while that of mild erosion showed a changing trend of first decreasing (2000–2005), then increasing (2005–2015), and finally decreasing (2015–2019). The area of moderate and above erosion showed a decreasing trend (2000–2019). In terms of total erosion, the proportion of total erosion accounted for by mild, intensive, and extreme erosion first increased (2000–2005), then decreased (2005–2015), and finally increased (2015–2019), while

that of total erosion accounted for by severe erosion first decreased (2000–2005), then increased (2005–2010) and finally decreased (2010–2019). The above analysis shows that from 2000 to 2019, the soil erosion intensity in the Wenchuan earthquake-stricken area showed a spatial change pattern of "overall stability and local intensification.

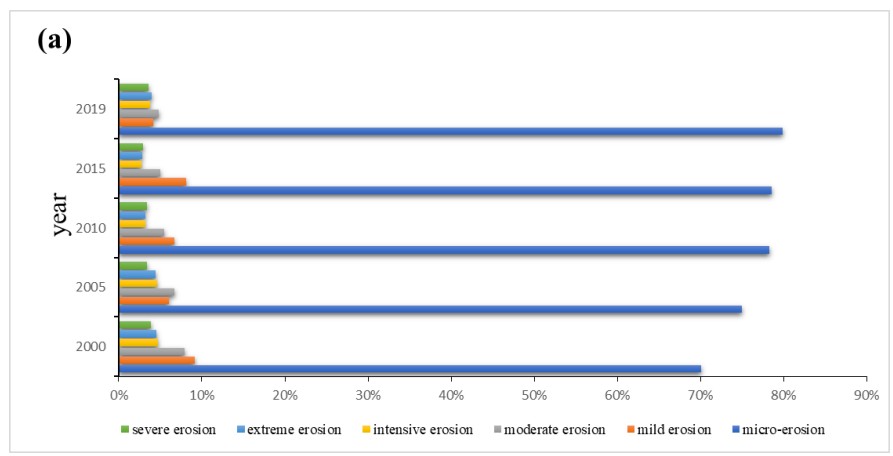

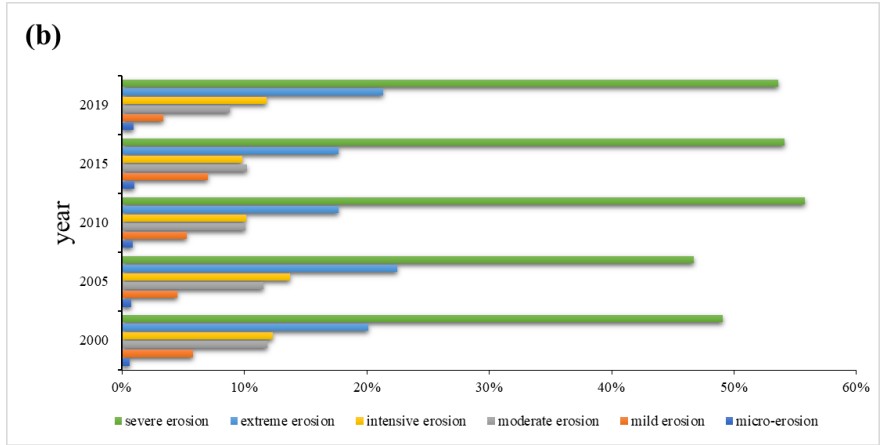

**Figure 3.** Ratio of erosion area and erosion amount in different years from 2000 to 2019. (**a**) Comparison of areas with different erosion levels (**b**) Comparison of erosion amounts from different erosion levels.

### 3.1.1. Spatial Distribution of Soil Erosion in Different Slope Zones

The slope greatly affects the soil erosion intensity and the amount of soil erosion [25]. According to the topography and geomorphology characteristics of the Wenchuan earthquake-stricken area, the slope was divided into five levels: <8°, 8°~15°, 15°~25°, 25°~35° and >45°, and the calculation results were shown in Table 3.

**Table 3.** Soil erosion amount and erosion area at different slope levels.

| Slope/(°) | Erosion Area/km² | Erosion Modulus/ (t·(km²·a) $^{-1}$) | Total Erosion/ (10,000 t·a $^{-1}$) | Area Ratio/% | Erosion Rate/% |
|---|---|---|---|---|---|
| 0~8 | 345,484 | 1826.55 | 631.04 | 70.63 | 61.68 |
| 8~15 | 97,852 | 2305.21 | 225.57 | 20 | 22.26 |
| 15~25 | 41,787 | 3262.76 | 136.34 | 8.54 | 13.56 |
| 25~35 | 3958 | 6167.41 | 24.41 | 0.81 | 2.44 |
| 35~45 | 66 | 10,359.52 | 0.68 | 0.01 | 0.07 |

Table 3 showed that the soil erosion modulus of the 35~45° zone was the largest at 10,359.52 t/(km²·a), belonging to the extremely intensive erosion type. The soil erosion mod-

ulus of the 25~35° zone was the second largest at 6167.41 t/(km²·a), which belongs to the intensive erosion type. The soil erosion modulus of the 15~25° zone was 3262.76 t/(km²·a), which belonged to the moderate erosion type. The soil erosion moduli of the 0~8° and 8~15° slope zones were 241,823.55 t/(km²·a) and 2305.21 t/(km²·a), respectively, which belonged to the mild erosion type. The proportion of soil erosion in >8° zones was larger than in zones with a slope of <8°. Therefore, soil erosion prevention and control should be strengthened in zones with slopes > 8° in the Wenchuan earthquake-stricken area.

As shown in Figure 4, in the same slope zone, different levels of soil erosion varied greatly, and micro-erosion accounted for the largest area proportion. The area proportion of micro-erosion in the 0~8° zone was more than 70%, while that in the 8~15° zone was more than 80%. It gradually decreased with an increase in slope. The area of severe erosion showed an increasing trend with an increase in the slope.

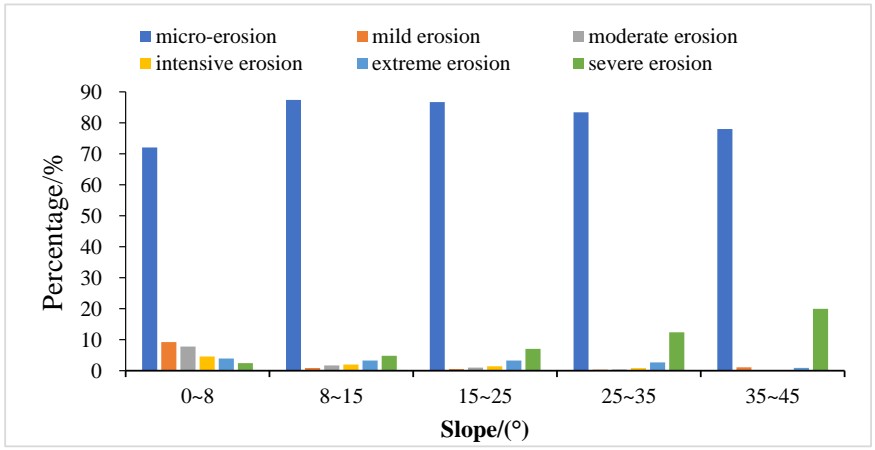

**Figure 4.** Percentage of soil erosion area at different gradients.

3.1.2. Spatial Distribution of Soil Erosion in Different Altitudes

According to the altitude distribution of the Wenchuan earthquake-stricken area, it was divided into six levels: <2000 m, 2000 m~3000 m, 3000 m~4000 m, 4000 m~5000 m, and >6000 m. It can be seen from Table 4 that the soil erosion modulus had a positive relationship with altitude, and the soil erosion modulus reached a maximum value of 39,429.6 t/(km²·a) in regions with altitude >6000 m. In regions with altitude >4000 m, the soil erosion modulus were 8863.43 t/(km²·a), 18,039.72 t/(km²·a) and 39,429.60 t/(km²·a), respectively. The main reason was that the bare land and barren grass were widely distributed in zones of altitude >4000 m, and the terrain in these areas was steep. After the Wenchuan earthquake in 2008, secondary disasters such as landslides, debris flows, and concentrated rainfall contributed to intensified soil erosion [26].

**Table 4.** Soil erosion amount and erosion area at different altitude zones.

| Altitude/m | Erosion Area/km² | Erosion Modulus/ (t·(km²·a)⁻¹) | Total Erosion/ (10,000 t·a⁻¹) | Area Ratio/% | Erosion Amount Ratio/% |
|---|---|---|---|---|---|
| <2000 | 304,670 | 1489.40 | 453.78 | 61.81 | 43.56 |
| 2000~3000 | 72,491 | 1736.07 | 125.85 | 14.71 | 12.13 |
| 3000~4000 | 80,296 | 1606.73 | 129.01 | 16.29 | 12.69 |
| 4000~5000 | 34,872 | 8863.43 | 309.08 | 7.07 | 30.52 |
| 5000~6000 | 594 | 18,039.72 | 10.72 | 0.12 | 1.06 |
| >6000 | 8 | 39,429.6 | 0.32 | 0.002 | 0.03 |

As shown in Figure 5, from the erosion area, the overall soil erosion area in the Wenchuan earthquake-stricken area from 2000 to 2019 first decreased, then increased, then decreased with an increase in altitude. The proportion of soil erosion area at an altitude

<2000 m was 61.81%, the proportion of soil erosion area at an altitude of 2000~3000 m was 14.71%, and the proportion of soil erosion area at an altitude of 3000~4000 m was 16.29%. However, the total erosion area in the region with an elevation ≥5000 m accounts for less than 2%. From the perspective of soil erosion, soil erosion in the Wenchuan earthquake-stricken area from 2000 to 2019 first decreased, then increased, and finally decreased with an increase in altitude. The proportion of soil erosion at an altitude <2000 m was 43.56%, and the proportion of soil erosion at an altitude of 2000~3000 m was 13.13%. The proportion of erosion at an altitude of 3000~4000 m was 12.69%, the proportion of erosion at an altitude of 4000~5000 m was significantly increased, accounting for 30.52%, while the sum of erosion at an altitude of ≥5000 m was less than 2%.

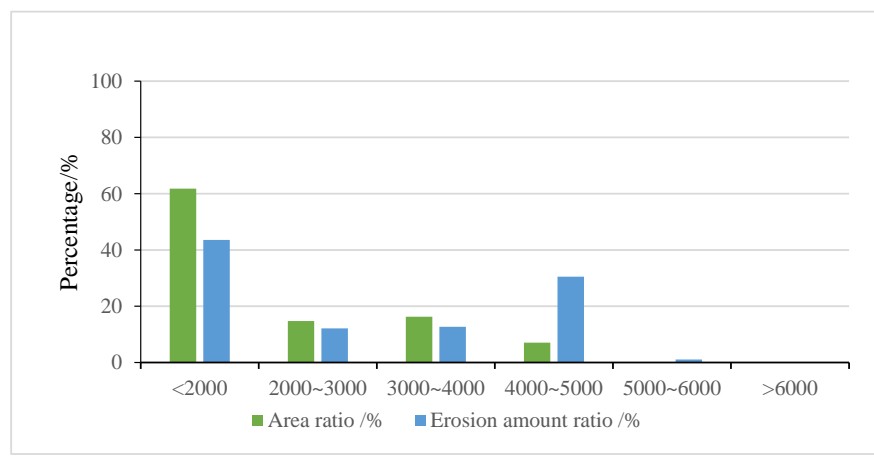

**Figure 5.** Proportion of erosion area and proportion of erosion amount at different altitudes.

### 3.1.3. Spatial Distribution of Soil Erosion in Different Land Use Types

Table 5 shows that the average soil erosion intensity index of dry land, meadow, and bare land in the Wenchuan earthquake-stricken area from 2000 to 2019 was 172.1, 205.51, and 301.56, respectively, indicating that soil erosion was relatively severe. The erosion intensity indexes of paddy fields, forested land, shrub land, and construction land were 120.77, 114.16, 130.31, and 121.52, respectively. The area of micro-erosion occupied the highest proportion of different land use types. Among them, the moderate and above erosion area of dry land and meadow accounted for nearly 30%, and the moderate and above erosion area of bare land accounted for more than 50%. Therefore, afforestation should be strengthened to reduce soil erosion [27].

**Table 5.** Erosion intensity index of different land use types and area percentage of erosion grade.

| Land Use Type | Micro-Erosion/% | Mild Erosion/% | Moderate Erosion/% | Intensive Erosion/% | Extreme Erosion/% | Severe Erosion/% | Soil Intensity Index |
|---|---|---|---|---|---|---|---|
| Paddy fields | 86.15 | 9.62 | 2.49 | 0.97 | 0.59 | 0.18 | 120.77 |
| Dry land | 62.94 | 16.44 | 11.53 | 4.88 | 3.09 | 1.12 | 172.10 |
| Forested land | 96.49 | 0.14 | 0.40 | 0.45 | 0.75 | 1.78 | 114.16 |
| Shrub land | 92.09 | 0.45 | 1.09 | 1.20 | 1.79 | 3.38 | 130.31 |
| Meadow | 66.22 | 4.98 | 7.70 | 6.42 | 7.51 | 7.17 | 205.51 |
| Bare land | 27.54 | 13.13 | 23.74 | 17.56 | 13.16 | 6.76 | 301.56 |
| Construction land | 93.50 | 0.75 | 1.02 | 1.82 | 1.31 | 1.61 | 121.52 |

The comparative analysis results of the soil erosion intensity index are shown in Figure 6. As shown in Figure 6a, the soil erosion intensity index of construction land increased from 105.9 to 116.28 during 2000–2005, an increase of 10.32, indicating that the soil erosion intensity of construction land increased. The soil erosion intensity indices of paddy fields, dry land, forested land, shrub land, meadow, and bare land decreased by 9.86, 7.71, 6.92, 10.6, 8.64, and 5.6, respectively. As shown in Figure 6b, the soil erosion

intensity indexes of forested land, shrub land, and construction land increased by 7.57, 4.45, and 6.56, respectively, from 2005 to 2010, while the soil erosion intensity indexes of paddy fields, dry land, meadow, and bare land decreased by 6.18, 22.96, 15.14, and 65.66, respectively. The decrease in dry land and bare land was the largest, indicating that although the proportion of high erosion intensity area in bare land increased, the overall soil erosion intensity showed a decreasing trend. As shown in Figure 6c, the soil erosion intensity indexes of paddy fields and construction land increased by 4.95 and 4.24, respectively, from 2010 to 2015, while the soil erosion intensity indexes of dry land, forested land, shrub land, meadow, and bare land decreased by 4.31, 9.67, 8.08, 5.06, and 41.83, respectively. The decrease in bare land was the largest, indicating that the overall soil erosion intensity showed a downward trend. As shown in Figure 6d, the soil erosion intensity index of paddy fields, dry land, forested land, shrub land, meadow, bare land, and construction land increased by 2.72, 12.27, 4.38, 0.17, 9.43, 40.65, and 8.37, respectively, from 2015 to 2019, and the overall erosion intensity increased. Mainly related to geological disasters and human deforestation and reclamation, after the earthquake, the rock structure was damaged, the soil was loose, the vegetation disappeared, and the soil erosion in the stricken area was intensified by heavy rain.

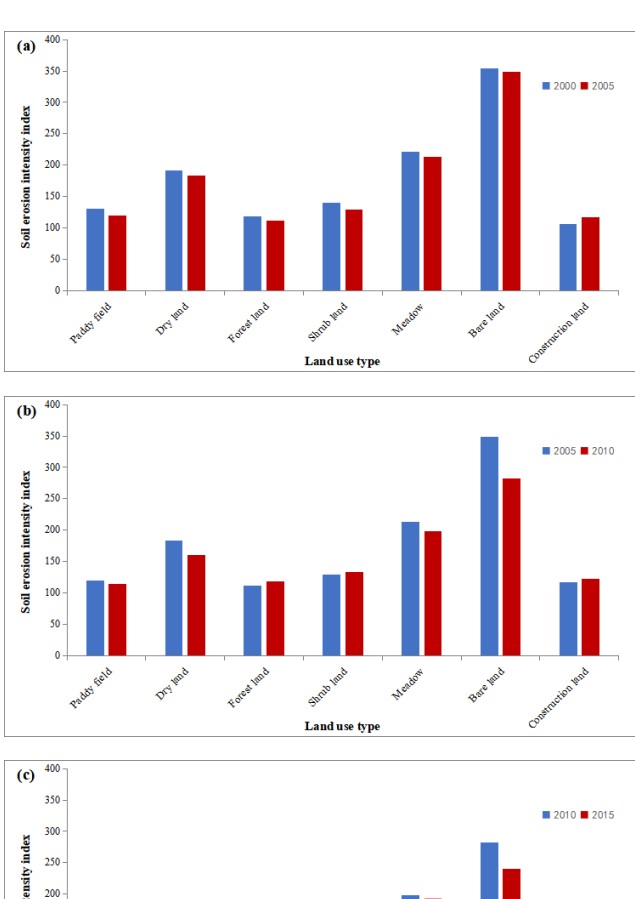

**Figure 6.** *Cont.*

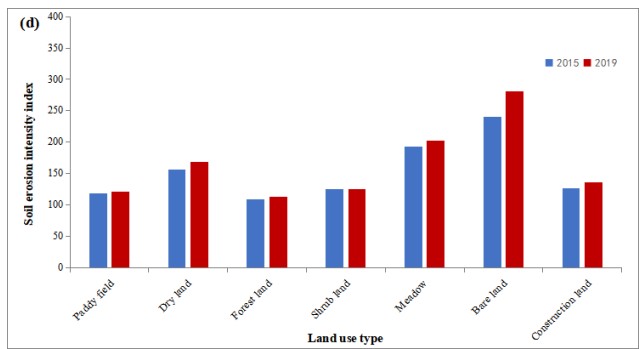

**Figure 6.** Change in soil erosion intensity index of different land use types. (**a**) 2000–2005 (**b**) 2005–2010 (**c**) 2010–2015 (**d**) 2015–2019.

3.1.4. Spatial Distribution of Soil Erosion in Different Levels of Earthquake Risk Areas

Based on different earthquake risk levels, the erosion modulus and amount of soil erosion were calculated in 2005 and 2010 (before and after earthquakes). The results are shown in Tables 6 and 7. The risk levels were divided into Levels 1, 2, 3, 4, and 5, from weak to strong. In the Sichuan earthquake area in 2005, the total erosion amount was $5.90 \times 10^8$ t/a, while in 2010, the total erosion amount increased to $6.25 \times 10^8$ t/a. Zones of earthquake risk Level 2 were the most widely distributed areas, accounting for 44.78%, followed by earthquake risk Level 1, accounting for 35.65%. In terms of the erosion amount, zones of earthquake risk Level 2 had the largest proportion of erosion amount, which was 54.27% and 53.57%, followed by earthquake risk Level 3, which was 23.13% and 33.05%.

**Table 6.** Soil erosion amount and erosion area of different earthquake risk levels (2005).

| Earthquake Risk | Erosion Area/km$^2$ | Erosion Modulus/(t·(km$^2$·a) $^{-1}$) | Total Erosion/ (10,000 t·a $^{-1}$) | Area Ratio/% | Erosion Rate/% |
|---|---|---|---|---|---|
| Level 1 | 103,861.44 | 976.26 | 10,139.54 | 35.65 | 17.17 |
| Level 2 | 130,468.86 | 2456.80 | 32,053.56 | 44.78 | 54.27 |
| Level 3 | 52,065.33 | 2623.44 | 13,659.03 | 17.87 | 23.13 |
| Level 4 | 4218.46 | 5900.32 | 2489.03 | 1.45 | 4.21 |
| Level 5 | 717.41 | 10,072.34 | 722.60 | 0.25 | 1.22 |
| Total | 291,332 | —— | 59,063.76 | 100 | 100 |

Note: —— refers to null.

**Table 7.** Soil erosion amount and erosion area of different earthquake risk levels (2010).

| Earthquake Risk | Erosion Area/km$^2$ | Erosion Modulus/(t·(km$^2$·a) $^{-1}$) | Total Erosion/ (10,000 t·a $^{-1}$) | Area Ratio/% | Erosion Rate/% |
|---|---|---|---|---|---|
| Level 1 | 103,862.56 | 529.35 | 5497.95 | 35.65 | 8.80 |
| Level 2 | 130,468.48 | 2565.39 | 33,470.24 | 44.78 | 53.57 |
| Level 3 | 52,065.36 | 3965.15 | 20,644.71 | 17.87 | 33.05 |
| Level 4 | 4218.49 | 5098.36 | 2150.74 | 1.45 | 3.44 |
| Level 5 | 717.40 | 9901.94 | 710.36 | 0.25 | 1.14 |
| Total | 291,332 | —— | 62,474.00 | 100 | 100 |

Note: —— refers to null.

*3.2. Soil Erosion Change Intensity in the Wenchuan Earthquake Area from 2000 to 2019*

3.2.1. Area Transfer of Soil Erosion Intensity of Different Grades

In order to analyze the changes in soil erosion grades over different years, the transfer of areas between different grades of soil erosion intensity was analyzed. Figure 7 shows that from 2000 to 2005, the conversion of mild erosion areas to moderate erosion areas was the largest (9980.69 km$^2$), followed by moderate erosion to intensive erosion areas (9242.22 km$^2$). From 2005 to 2010, the conversion of moderate erosion areas to light erosion

areas was the largest (9545.39 km$^2$), followed by intensive erosion to moderate erosion areas (8111.86 km$^2$). From 2010 to 2015, the conversion of mild erosion areas to mild erosion areas was the largest (7496.66 km$^2$), followed by moderate erosion to mild erosion areas (6953.24 km$^2$). From 2015 to 2019, the conversion of mild erosion areas to mild erosion areas was the largest (8418.88 km$^2$), followed by mild erosion to moderate erosion areas (6894.54 km$^2$).

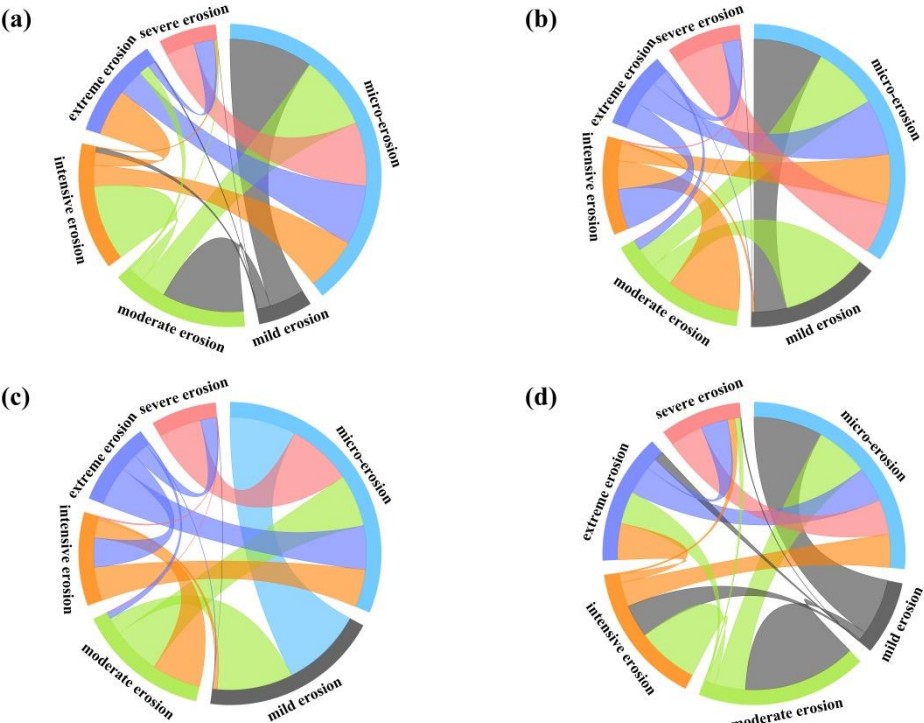

**Figure 7.** Area transfer matrix among different erosion levels. (**a**) 2000–2005 (**b**) 2005–2010 (**c**) 2010–2015 (**d**) 2015–2019.

3.2.2. Gravity Center Mitigation Trajectory of Soil Erosion Intensity

The distribution and migration of the soil erosion gravity center could effectively reflect the change rate and increment of soil erosion in the study area. In this study, MATLAB 2023 and ArcGIS 10.7 were used to calculate and analyze the distribution and change in the gravity center of soil erosion intensity in the preceding 20 years (Figure 8). In the preceding 20 years, the soil erosion gravity centers were mainly distributed in the central part of the Wenchuan earthquake area. In order to analyze the spatial change pattern of soil erosion in the Wenchuan earthquake-stricken area in the preceding 20 years, this study analyzed the migration direction of the gravity center. The results showed that the gravity center of soil erosion shifted from Jiangyou City to Pingwu County in the northwest of Sichuan Province from 2000 to 2005, indicating that the increment and growth rate of soil erosion in the northwest of the Wenchuan earthquake area was greater than that in the southeast. The soil erosion center shifted from Pingwu County to Maoxian County in the southwest from 2005 to 2015, indicating that the increment and growth rate of soil erosion in the southwest of the Wenchuan earthquake area was greater than that in the northeast.

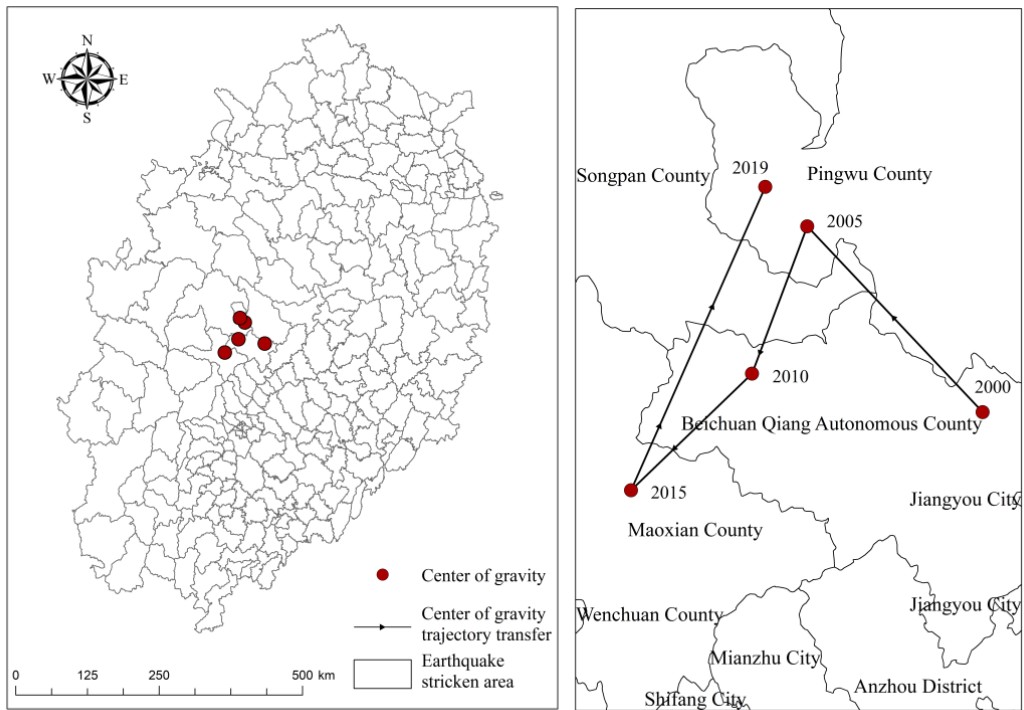

**Figure 8.** Spatial distribution and mitigation trajectory of soil erosion gravity center from 2000 to 2019.

*3.3. Dominant Factor of Soil Erosion Change Wenchuan Earthquake Stricken Area in Different Historical Periods*

### 3.3.1. Single Factor

The contribution rate of various factors in different periods on soil erosion in the Wenchuan earthquake-stricken area was calculated using Geodetector. The results (Table 8 and Figure 9) showed that the q values of different factors in the Wenchuan earthquake-stricken area in 2000 were ranked as follows: vegetation > terrain > land use > air temperature > population density > slope > precipitation > GDP. Vegetation, landform, and land use type had the largest explanatory power among all factors, with q values of 0.612, 0.240, and 0.180, respectively. On the contrary, temperature, population density, slope, precipitation, and GDP had less influence, with q values less than 0.1, which were 0.092, 0.032, 0.028, 0.016, and 0.012, respectively.

**Table 8.** q values of different single factor q values from 2000 to 2019.

| Factor \ Time | 2000 | 2005 | 2010 | 2015 | 2019 |
|---|---|---|---|---|---|
| Landform | 0.168 | 0.283 | 0.165 | 0.3 | 0.294 |
| Slope | 0.02 | 0.039 | 0.028 | 0.025 | 0.02 |
| Land use | 0.126 | 0.269 | 0.098 | 0.123 | 0.168 |
| GDP | 0.008 | 0 | 0.011 | 0.011 | 0.011 |
| Population density | 0.022 | 0.076 | 0.048 | 0.062 | 0.003 |
| Temperature | 0.064 | 0.092 | 0.067 | 0.09 | 0.095 |
| Precipitation | 0.011 | 0.176 | 0.022 | 0.042 | 0.05 |
| Vegetation | 0.428 | 0.554 | 0.372 | 0.356 | 0.484 |

As shown in Figure 9, the q values of influencing factors in the Wenchuan earthquake-stricken area in 2000 were ranked as follows: vegetation (0.428) > terrain (0.168) > land use (0.126) > air temperature (0.064) > population density (0.022) > slope (0.020) > precipitation (0.011) > GDP (0.008). The q values in 2005 were as ranked as follows: vegetation (0.554) > terrain (0.283) > land use (0.269) > precipitation (0.176) > air temperature (0.092) > population density (0.076) > slope (0.039) > GDP (0). The q values of soil erosion

factors in the Wenchuan earthquake-stricken area in 2010 were ranked as follows: vegetation (0.372) > terrain (0.165) > land use (0.098) > air temperature (0.067) > population density (0.048) > slope (0.028) > precipitation (0.022) > GDP (0.011). The q values of soil erosion factors in the Wenchuan earthquake-stricken area in 2015 were ranked as follows: vegetation (0.356) > terrain (0.300) > land use (0.123) > air temperature (0.090) > population density (0.062) > precipitation (0.042) > slope (0.025) > GDP (0.011). The q values of soil erosion factors in the Wenchuan earthquake-stricken area in 2019 were ranked as follows: vegetation (0.484) > terrain (0.294) > land use (0.168) > air temperature (0.095) > precipitation (0.050) > slope (0.020) > GDP (0.011) > population density (0.003). In terms of the explanatory power of natural factors on soil erosion in earthquake-stricken areas, the q value of vegetation was the largest, followed by that of terrain. Temperature, slope, and precipitation also had fewer effects on soil erosion, with q values < 0.3. The q value of land use was the largest among human factors, which was 0.269 and 0.168 in 2005 and 2019, respectively. Population density and GDP have fewer effects.

As shown in Figure 9, the explanatory powers of topography, slope, and GDP were relatively unchanged from 2000 to 2019. The q value of precipitation increased from 0.011 in 2000 to 0.176 in 2005, second only to the explanatory power of land use type, and then dropped to 0.022 in 2010. The explanatory power in 2015 and 2019 was larger than the slope. Population density and GDP had fewer impacts on soil erosion. The q value of vegetation increased first (2000–2005), then decreased (2005–2010), and finally increased (2010–2019). It could be seen that the explanatory power of vegetation, precipitation, and land use type changed greatly between 2000 and 2019.

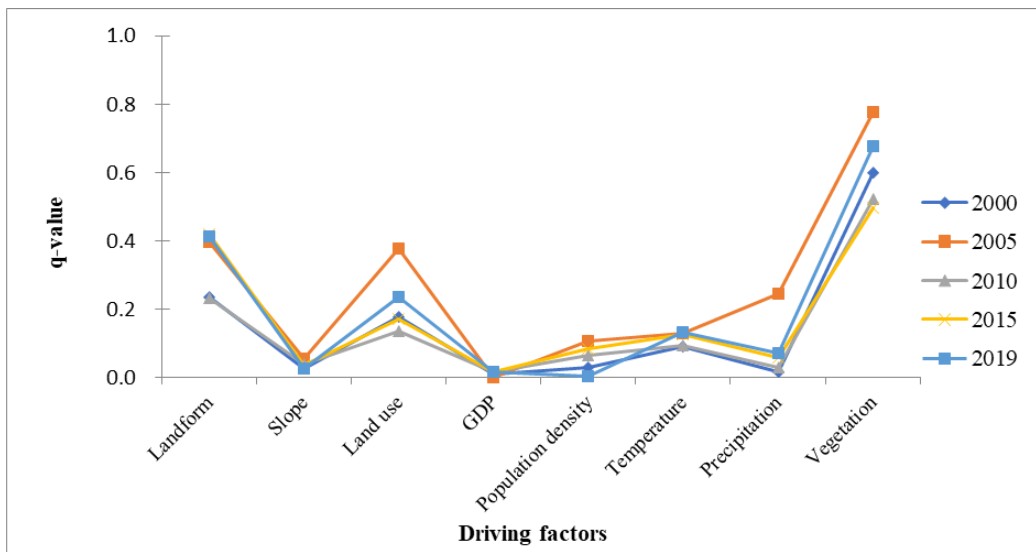

**Figure 9.** Dominant single factor in different years.

### 3.3.2. Interactive Factor

The interactions between two factors were obtained using interactive detection (Figure 10). In 2000, the vegetation ∩ landform, vegetation ∩ slope, and vegetation ∩ land use had larger explanatory power with q values of 0.707, 0.768, and 0.787, respectively. The GDP ∩ population intensity had the smallest q value of 0.041. In 2005, the vegetation ∩ landform, vegetation ∩ slope, and vegetation ∩ land use had larger explanatory power with the q values of 0.839, 0.832, and 0.938, respectively. The GDP ∩ slope had the smallest q value of 0.056. In 2010, the vegetation ∩ landform, vegetation ∩ slope, and vegetation ∩ land use had larger explanatory power with q values of 0.699, 0.934, and 0.798, respectively. The GDP ∩ precipitation had the smallest q value of 0.052. In 2015, the vegetation ∩ landform, vegetation ∩ slope, and vegetation ∩ land use had larger explanatory power with q values of 0.760, 0.769, and 0.816, respectively. The GDP ∩ slope had the smallest q value of 0.039. In 2019, the vegetation ∩ precipitation, vegetation ∩ slope, and vegetation ∩ land use

had larger explanatory power with q values of 0.851, 0.845, and 0.979, respectively. The population density ∩ slope had the smallest q value of 0.038.

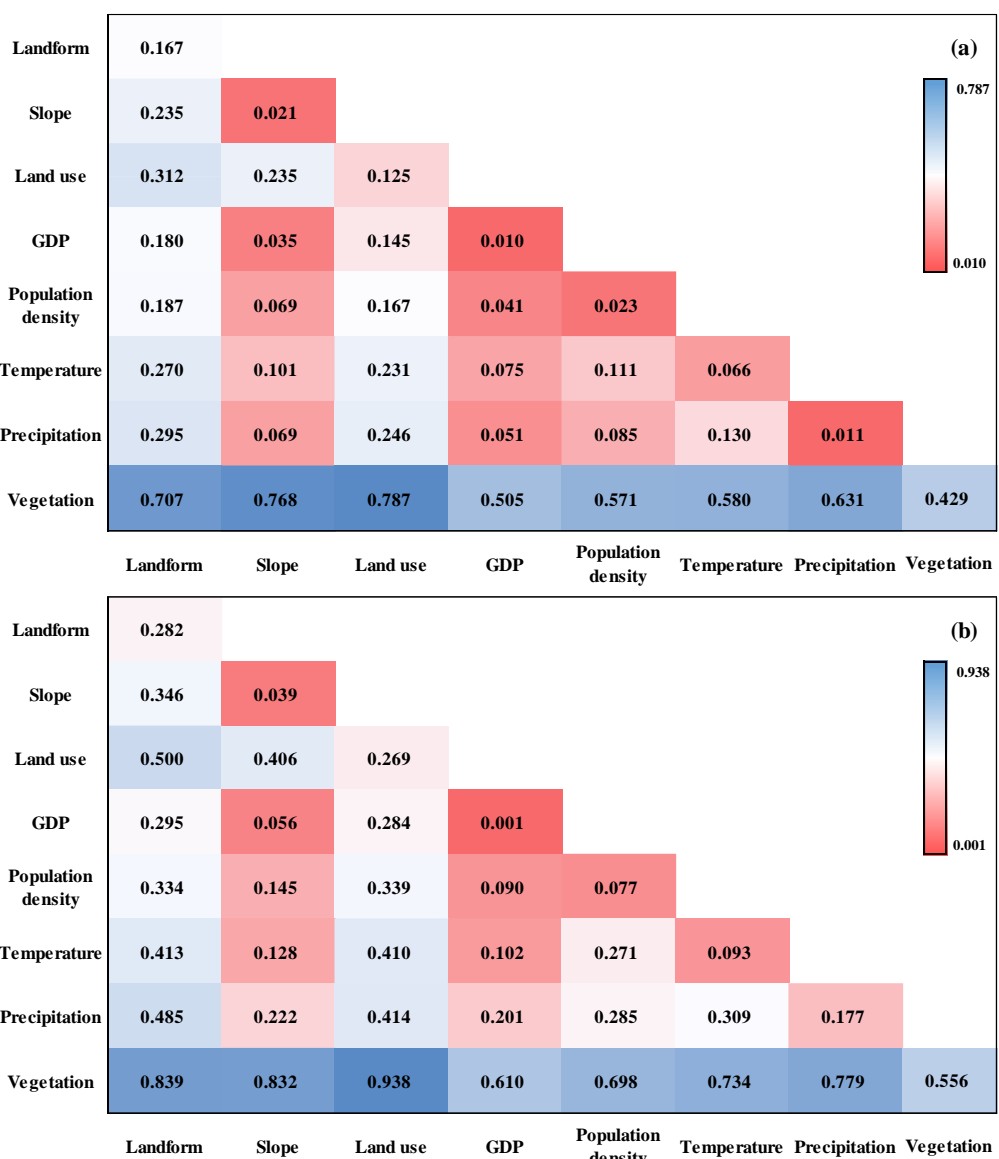

**Figure 10.** *Cont.*

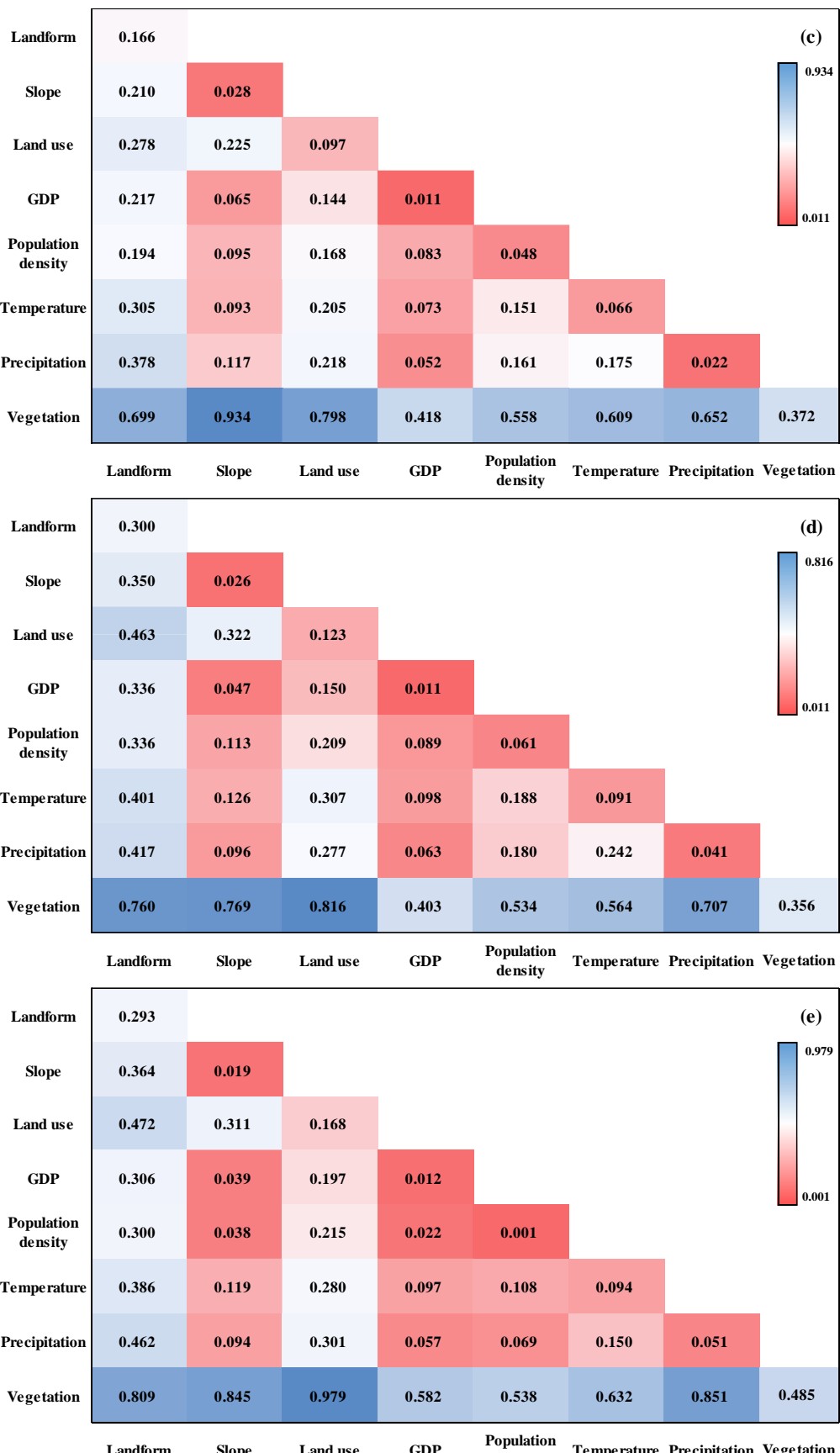

**Figure 10.** Dominant interactive factors in different periods. (**a**) 2000 (**b**) 2005 (**c**) 2010 (**d**) 2015 (**e**) 2019.

## 4. Discussion

### 4.1. Causes of Spatial Distribution Pattern of Soil Erosion in Wenchuan Earthquake Stricken Area

During 2000–2005, the overall erosion intensity was lower than that in 2000. Micro-erosion was mainly distributed in the middle of the Jialing River Basin, the southern part of the Mintuo River Basin, and the western part of the Hanjiang River Basin, such as Wenxian County, Wenchuan County, Pingwu County, Baoxing County, and other places. The reason was that these areas had large areas of shrubbery, forested land, and grassland. Intensive and severe erosion were mainly distributed in the northern part of the Mintuo River Basin and the eastern part of the Shigu River Basin in the Jinsha River, such as Lixian, Xiaojin, Luding, and Jiulong counties, because of the widely distributed land use of bare land and low-cover grassland in these areas [28].

During 2005–2020, the erosion intensity increased significantly, and the spatial scope of intensive erosion showed an obvious expansionary trend. It was mainly because of secondary disasters such as landslides and debris flows caused by the Wenchuan earthquake in 2008, which greatly destroyed the local vegetation ecosystem, especially in Lixian, Wenchuan, Xiaojin, and other areas near the Longmenshan fault zone, which led to a large area of extreme and severe erosion [29].

During 2010–2015, the overall soil erosion intensity showed a stable trend. In 2015, the erosion intensity of the northern part of the Mintuo River Basin, the eastern part of the Jinsha River Basin increased significantly, mainly due to the significant increase in rainfall erosivity in this area, from 800 $(MJ \cdot mm)/(hm^2 \cdot h \cdot a)$ to 1400 $(MJ \cdot mm)/(hm^2 \cdot h \cdot a)$, which aggravated the soil erosion in this area. However, due to the decrease in rainfall erosivity, the erosion intensity of Zhenyuan County, Kongtong District, Jingchuan County, and Lingtai County in the west of the Longmen–Sanmenxia Basin decreased significantly [30].

During 2015–2019, the erosion area of micro-erosion, intensive erosion, extreme erosion, and severe erosion increased, respectively, and the area of other erosion grades decreased to a lesser extent. The ratio of erosion amount attributable to intensive erosion and extreme erosion increased, respectively, while the erosion amount of other erosion grades showed a downward trend. It could be seen that the soil erosion intensity increased in 2019. In 2019, the erosion intensity of Aba County and Maerkang City in the northern part of the Mintuo River Basin decreased, mainly due to a reduction in the area covered by damaged shrubs and grasslands in the region and the erosion intensity decreased [31]. The rainfall erosivity in 2019 increased by 437 $(MJ \cdot mm)/(hm^2 \cdot h \cdot a)$ compared with that in 2015, resulting in a significant increase in erosion intensity in Jingning County, Zhuanglang County, Qin'an County, Pengyang County, Zhenyuan County, and Kongtong District in the west of the Longmen–Sanmenxia Basin.

### 4.2. Effects of Earthquake Hazards on Temporal and Spatial Changes in Soil Erosion

In 2005, compared with 2000, the erosion rate of earthquake risk Level 2 increased by 8.4%, while that of earthquake risk Level 1 decreased by 10.21%. In 2010, the erosion rate of earthquake risk Level 3 increased by 9.92%, while that of earthquake risk Level 1 decreased by 8.37%. The erosion amount of other risk levels decreased slightly, and the erosion amount of high-level earthquake risk increased overall [32]. In 2010, compared with 2005, the erosion area of severe erosion increased by 59.64 $km^2$, and the amount of erosion increased by 500, 24,600 t. The Wenchuan earthquake in 2008 destroyed a large area of vegetation, leading to the worsening of soil erosion [33]. In 2015, the soil erosion in earthquake risk Level 1 increased by 5.06%, while that of earthquake risk Level 3 decreased by 6.69%. In 2015, compared with 2010, the average soil erosion modulus decreased by 223.78 $t/(km^2 \cdot a)$, and the total erosion decreased by 110.3555 million t. It could be seen that the impact of the earthquake was weakened after 10 years. In 2019, the erosion amount of earthquake risk Levels 1 and 2 increased by 2.03% and 1.35%, respectively, while the erosion amount of other risk levels decreased slightly [34]. Overall, the earthquake could have caused secondary hazards, such as landslides and debris flows, which would destroy the surface vegetation ecosystems and finally aggravate the erosion intensity. In addition,

the earthquakes would make the soil structure loose, which was conducive to soil and water loss.

*4.3. Effects of Human Activities on Temporal and Spatial Changes in Soil Erosion*

The effects of human activities on the temporal and spatial changes in soil erosion were mainly reflected in the changes in land use types. The influence of population density on soil erosion fluctuated slightly from 2005 to 2015 but was extremely weak in 2019. Land use type was an important factor that can be controlled by human beings to affect soil erosion, and its single factor influence was second only to vegetation and terrain [35]. In 1999, Sichuan, Shaanxi, and Gansu provinces took the lead in pilot projects to return farmland to forests, and in 2002, the country launched a project to return farmland to forests. The soil erosion moduli in 2000 and 2005 were 2397.04 t/(km$^2$·a) and 2111.40 t/(km$^2$·a), respectively. Compared with 2000, the average soil erosion modulus in 2005 was reduced by 285.64 t/(km$^2$·a), and the total erosion was also reduced. It could be seen that soil erosion in 2005 was reduced due to the implementation of the policy of returning farmland to forest. Soil erosion increased significantly in 2010 compared to 2005. From 2005 to 2010, the soil erosion index of forested land, shrub land, and construction land increased by 7.57, 4.45, and 6.56, respectively, while the soil erosion index of other land use types decreased. There was no significant change in land use type, and human activities had little effect on soil erosion, mainly due to the secondary disasters after the earthquake. From 2015 to 2019, the area of damaged meadow cover increased, and the erosion intensity decreased in some areas of the northern Bintuo River basin [36,37].

**5. Conclusions**

In this study, we quantitatively analyzed the spatiotemporal variation characteristics of soil erosion in the Wenchuan earthquake-stricken area from 2000 to 2019 based on the RUSLE model and gravity center model and then determined its dominant factors in different periods using Geodetector. The main conclusions are as follows:

(1) From 2000 to 2019, the total erosion amount in the Wenchuan earthquake-stricken area was $10.05 \times 10^8$ t, and the average soil erosion modulus was 2038.2 t/(km$^2$·a), which belonged to mild erosion. Severe soil erosion area was mainly distributed in the northern part of the Mintuo River Basin, the lower reaches of the Jinsha River, and the middle section of the Longmen–Sanmenxia River Basin.

(2) The spatiotemporal change patterns of soil erosion were greatly affected by the slope. After the Wenchuan earthquake in 2008, the soil erosion intensity showed an increasing trend with the combined actions of secondary disasters and concentrated rainfall.

(3) Landslides, debris flows, and floods caused by the Wenchuan earthquake contributed to aggravating the soil erosion intensity in the stricken area.

(4) During 2000–2019, the soil erosion intensity showed an overall decreasing trend, while the soil erosion intensity showed an increasing trend around 2008 due to the Wenchuan earthquake.

(5) During 2000–2019, soil erosion in the Wenchuan earthquake-stricken area has been greatly affected by vegetation, terrain, and land use types.

These research results can provide important decision support for soil erosion control and ecosystem restoration in the Wenchuan earthquake-stricken area. In addition, these results would be conducive to revealing and understanding the interactive process of "Water–Soil–Vegetation" in mountainous regions all over the world.

**Author Contributions:** Conceptualization, methodology, writing—original draft preparation, J.L.; investigation, supervision, project administration, and funding acquisition, B.G. and G.Y.; Investigation, K.Y. All authors have read and agreed to the published version of the manuscript.

**Funding:** This research was funded by the Natural Science Foundation of Shandong Province, grant number ZR2021MD047; Scientific Innovation Project for Young Scientists in Shandong Provincial Universities, grant number 2022KJ224; National Natural Science Foundation of China, grant number 42101306 and 62006096; Natural Science Foundation of Fujian Province, grant number 2020J05146 and a grant from the State Key Laboratory of Resources and Environmental Information System.

**Institutional Review Board Statement:** Not applicable.

**Informed Consent Statement:** Not applicable.

**Data Availability Statement:** Not applicable.

**Conflicts of Interest:** The authors declare no conflict of interest.

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
