# Peer review of "The Spatiotemporal Variations in Soil Erosion and Its Dominant Influencing Factors in the Wenchuan Earthquake-Stricken Area"

_sustainability, doi:10.3390/su151712701_

Round 1
Reviewer 1 Report
It is a very interesting work. This research is of great significance to study the the spatio-temporal variatons of soil erosion and its dominant influencing factors by earthquakes based on the RUSLE model, gravity center model, and geographic detectors. The work is nicely presented, with a correct methodological approach, well-supported results and conclusions. However, I suggest revising the manuscript (i.e., minor revision) focusing mainly in the figures, methodology and results.

Reviewer 2 Report
The article is devoted to a very relevant topic and uses modern calculation equations to confirm the degree of erosion of the territory.
However, there are a few notes:
On fig. 2 in the legend, the lake is indicated, but in the figure the lake is not visible.
The conclusion should be made more concise and show the most important points of the study.
Accept in present form after minor revision.
Reviewer 3 Report
My comments are as follows:
1.Some more relative references should be added in the section of Introduction.
2. In the section of “2.1 Study area”, the 1500-3200m should be 1500~3200m.
3. In Table 1, intensity erosion should be intensive erosion, and some similar problems should be revised in other places
4. In Table 2, the Erosion Area/km2 should be Erosion Area/104 km2.
5. In 3.2.2 Gravity center mitigation trajectory of soil erosion intensity, the detail version of Matlab and ArcGIS should be added.
6. In Table 6 and 7, the 1,2,3…5 should be Level 1, Level 2, …..,Level 5.
7. The font type of Figure 5 should be revised.
8. Please delete the total value of Table 3, 4,5.
9. Some more recent references should be still added in the section of Discussions.
Reviewer 4 Report
This article introduces The Spatio-Temporal Variations of Soil Erosion and its Dominant Influencing Factors in Wenchuan Earthquake Stricken Area. The content of this article demonstrates a high level of quality in its writing and, with some necessary revisions, is suitable for acceptance in publications.
The abstract necessitates an update to offer a succinct overview of the study, encompassing the context, research query, hypothesis, methodology, key discoveries, and conclusions. Ideally, it should also contextualize these findings within the broader research domain.
The introduction section requires a more exhaustive deliberation leading to the articulation of the problem statement and the scope of the study. Furthermore, additional literature must be incorporated to substantiate the statements presented in the text. Please include the following citations: For lines (Under specific geological conditions, soil erosion can also cause geological disasters such as landslides, collapses, and debris flows. The resulting soil erosion will cause river siltation and aggravate flood disasters), https://doi.org/10.1016/j.enggeo.2022.106899; and for lines 118-119, https://doi.org/10.1016/j.trgeo.2021.100697.
The methodology necessitates refinement to ensure its logical coherence, clarity, and the potential for replication by fellow researchers. In the event that any segment or the entirety of the methodology has been previously published, a succinct summary along with appropriate citations is imperative.
Lastly, the conclusion must encapsulate all distinctive and significant findings of the study, accompanied by a clear delineation of the importance of the work.
moderate
Round 2
Reviewer 4 Report
Accepted
Minor